# Peripheral Blood Lymphocyte Phenotype Differentiates Secondary Antibody Deficiency in Rheumatic Disease from Primary Antibody Deficiency

**DOI:** 10.3390/jcm9041049

**Published:** 2020-04-07

**Authors:** Alexandra Jablonka, Haress Etemadi, Ignatius Ryan Adriawan, Diana Ernst, Roland Jacobs, Sabine Buyny, Torsten Witte, Reinhold Ernst Schmidt, Faranaz Atschekzei, Georgios Sogkas

**Affiliations:** Department of Rheumatology and Immunology, Hannover Medical School, 30625 Hannover, Germany; Jablonka.Alexandra@mh-hannover.de (A.J.); Etemadi.Haress@mh-hannover.de (H.E.); Adriawan.Ignatius@mh-hannover.de (I.R.A.); Ernst.Diana@mh-hannover.de (D.E.); Jacobs.Roland@mh-hannover.de (R.J.); Buyny.Sabine@mh-hannover.de (S.B.); Witte.Torsten@mh-hannover.de (T.W.); Reinhold.Ernst.Schmidt@mh-hannover.de (R.E.S.)

**Keywords:** hypogammaglobulinemia, secondary hypogammaglobulinemia, primary immunodeficiency, common variable immunodeficiency, methotrexate, DMARD, systemic lupus erythematosus, rheumatoid arthritis, CD4^+^ T follicular cells, class-switched memory B cells

## Abstract

The phenotype of primary immunodeficiency disorders (PID), and especially common variable immunodeficiency (CVID), may be dominated by symptoms of autoimmune disorders. Furthermore, autoimmunity may be the first manifestation of PID, frequently preceding infections and the diagnosis of hypogammaglobulinemia, which occurs later on. In this case, distinguishing PID from hypogammaglobulinemia secondary to anti-inflammatory treatment of autoimmunity may become challenging. The aim of this study was to evaluate the diagnostic accuracy of peripheral blood lymphocyte phenotyping in resolving the diagnostic dilemma between primary and secondary hypogammaglobulinemia. Comparison of B and T cell subsets from patients with PID and patients with rheumatic disease, who developed hypogammaglobulinemia as a consequence of anti-inflammatory regimes, revealed significant differences in proportion of naïve B cells, class-switched memory B cells and CD21^low^ B cells among B cells as well as in CD4^+^ memory T cells and CD4^+^ T follicular cells among CD4^+^ T cells. Identified differences in B cell and T cell subsets, and especially in the proportion of class-switched memory B cells and CD4^+^ T follicular cells, display a considerable diagnostic efficacy in distinguishing PID from secondary hypogammaglobulinemia due to anti-inflammatory regimens for rheumatic disease.

## 1. Introduction

Hypogammaglobulinemia as a consequence of hematological malignancies, systemic disorders causing excessive loss or catabolism of immunoglobulins, viral infections or drugs, such as antiepileptic agents and anti-inflammatory medications, is defined as secondary hypogammaglobulinemia [1,2]. Its exclusion is required for the diagnosis of primary immunodeficiency disorders (PID) [3]. Common variable immunodeficiency (CVID) is the most common form of symptomatic PID [4,5]. Besides bacterial infections due to primary antibody failure, CVID may manifest with autoimmunity, granulomatous and/or lymphoproliferative disease [6]. Autoimmunity in CVID includes rheumatic disease, such as rheumatoid arthritis (RA), systemic lupus erythematosus (SLE) and Sjögren’s syndrome (SS) [7,8,9]. These conditions necessitate anti-inflammatory treatment, which could lead to hypogammaglobulinemia even in the absence of a PID.

Secondary hypogammaglobulinemia is distinguished from primary hypogammaglobulinemia on the basis of a patient’s medical history [2,3]. In other words, the diagnosis of a PID is clear in case the diagnosis of hypogammaglobulinemia precedes immunosuppressive treatment of autoimmune or lymphoproliferative conditions. However, autoimmunity or lymphoproliferative disease and introduction of immunosuppressive treatment could precede the diagnosis of hypogammaglobulinemia [8,9,10,11,12]. The relatively recent discovery of monogenic disorders manifesting as CVID and subsequent studies of cohorts of patients with common monogenic defects revealed that hypogammaglobulinemia in primary immunodeficiency may have a later onset than autoimmunity or lymphoproliferative disease, which can phenotypically prevail [13,14,15,16]. In that case, distinction between primary and secondary hypogammaglobulinemia may become a diagnostic challenge.

After activating antigen recognition, progressive differentiation of T cells can be traced by characterizing CD45RA, CD27 and CD28 expression [17]. Upon activation, CD45RA switches to CD45RO, while CD28 and CD27 expression are lost sequentially in the course of T cell differentiation. In the case of CD8^+^ T cell differentiation, CD27^−^CD28^−^ cells (late effector cells) display an effector-like phenotype, whereas CD27^+^CD28^−^ cells (early effector cells) appear to have a recent replicative history and partial effector function [18]. Co-expression of CD31 and CD45RA defines recent thymic emigrants, associating with the presence of T cell receptor excision circles (TRECs) [19]. Follicular T cells are antigen-experienced CD4^+^ T cells, expressing CXCR5. These cells regulate antigen-specific activation of B cells in the context of a germinal center reaction but are also identified in peripheral blood [20]. After initial differentiation of B cells in the bone marrow, including the successful expression of a B cell receptor, transitional B cells emigrate from the bone marrow [21]. Expression of high levels of IgM and CD38 are used to characterize these cells. Naïve B cells are differentiated from memory ones on the basis of expression of CD27 and IgD [22]. CD21^low^ B-cells are an innate-like memory B cell subset, found to be increased in autoimmune diseases such as SLE and RA as well as in CVID patients with autoimmune manifestations [23,24,25]. Peripheral blood lymphocyte phenotyping has been traditionally employed to evaluate immunodeficiency within PID and define disease subgroups [25]. According to the current diagnostic criteria of CVID, characterization of particular B and T cell subsets is required for the diagnosis of CVID and its differentiation from combined immunodeficiency (CID) [3].

In a previous study, secondary antibody deficiency as a consequence of glucocorticoid therapy with or without methotrexate has been associated with reduced naïve and transitional B cells as well as with reduced CD4 memory T cells [26]. Class-switched memory cells, which are found to be reduced in the majority of CVID patients, remained on the other hand unaffected [26,27]. Additional changes in peripheral lymphocyte subset counts, such as an increased numbers of transitional and CD21^low^ B cells as well as elevated circulating CD4^+^ T follicular helper cells have been described in CVID [23,28,29]. So far, it remains unclear whether CVID-associated changes in peripheral lymphocyte counts could aid diagnostic distinction between primary and secondary hypogammaglobulinemia. Hence, we conducted a case-control study to examine the differences in peripheral B- and T- subpopulations between patients with secondary hypogammaglobulinemia due to immunosuppressive treatment of rheumatic disease and patients with CVID. Considering the identified differences, we propose criteria for the distinction of primary from secondary hypogammaglobulinemia and evaluate their diagnostic accuracy.

## 2. Experimental Section

This single-center study included all rheumatic disease patients with secondary hypogammaglobulinemia visiting our Rheumatology outpatient clinic between December 2018 and April 2019. Visits of patients were scheduled approximately every 3 to 6 months and serum immunoglobulin (Ig) levels were measured at every visit. Secondary hypogammaglobulinemia has been defined as persistently reduced IgG (< 7 g/L) at time of the study and in follow-up visits during at least the year before the study, developing as a consequence of anti-inflammatory regimens including prednisolone, diverse disease-modifying anti-rheumatic drugs (DMARD) and biological therapies [30]. The systemic lupus erythematosus disease activity index 2000 (SLEDAI-2K) and the disease activity score 28-c reactive protein score (DAS28-CRP) were calculated for SLE and RA patients, respectively, as described by others [31,32].

Patients with secondary hypogammaglobulinemia were compared with PID patients having an appointment in the Clinical Immunology outpatient clinic of Hannover Medical University at the same time. PID has been diagnosed in the absence of evidence for secondary hypogammaglobulinemia, including a history of previous steroid treatment and classified according to the current ESID diagnostic criteria [3]. Results of phenotypic analysis of lymphocytes from patients with secondary hypogammaglobulinemia were compared with data from patients with PID. Patients’ demographic data and Ig values are shown in Table 1. Table 2 shows the rheumatic diseases of patients with secondary hypogammaglobulinemia, their current treatment, the treatment at the time point of diagnosis of hypogammaglobulinemia as well as all previous anti-inflammatory treatments.

Phenotypic analysis of lymphocytes has been performed as described previously [33]. Briefly, peripheral blood mononuclear cells (PBMC) were obtained from heparinized blood samples of consenting patients by centrifugation over a Ficoll-Hypaque gradient. Phenotypic analyses were performed as multicolor immunofluorescence of PBMC, utilizing directly labeled monoclonal antibodies. 1 × 10^5^ to 2 × 10^6^ cells/well were incubated with murine monoclonal antibodies against the appropriate antigens at an optimal dilution for 20 min at 4° C. Nonspecific binding was eliminated by mixing the samples with a 1:5 solution of a commercial human IgG (Octagam; Octapharma, Lingolsheim, Germany). Samples were washed three times in phosphate buffered saline (PBS) supplemented with 0.5% bovine serum albumin (BSA) and at least 10^4^ cells per appropriate gate were analyzed using a FACSCanto II flow cytometer with Cell Quest software (Becton Dickinson, Franklin Lakes, USA). Offline data analysis was performed by using FCS Express software V6 (Denovo Software, Pasadena, USA). Gating strategy to measure B cell and T cell subsets is shown in Appendix A and Appendix A, respectively.

The following antibodies (all obtained from Biolegend, San Diego, USA, if not otherwise stated) were used for this study: CD3 PerCP (BD Pharmingen, Franklin Lakes, USA), CD3 PE, CD3 APC, CD4 FITC (Beckman Coulter, Brea, USA), CD4 PerCP, CD8 PE, CD8 PECy7, CD16 FITC, CD19 BV510, CD21 PE, CD24 FITC, CD27 FITC, CD28 BV421, CD31 FITC, CD38 PECy7, CD45R0 BV421, CD45RA BV510, CXCR5 PE, IgD PE, IgM Alexa Fluor 647. Each flow cytometric analysis was controlled with appropriate isotype-matched antibodies.

Differences between patients with primary and secondary hypogammaglobulinemia were evaluated with the Mann–Whitney test. Comparison of more than two groups was performed with the Kruskal–Wallis test. Correlation was evaluated with Spearman’s rho (r) analysis. For statistical calculation and diagrams we used GraphPad prism 5.00 (GraphPad, La Jolla, USA). Diagnostic value of studied lymphocyte subsets was evaluated with the “pROC” statistical package on R (v.3.6.0) [34].

This study was conducted in accordance with the Declaration of Helsinki and was approved from the Ethic committee of the Hannover Medical School (approval number: 8875 (21.03.14)). All patients signed an informed consent form.

## 3. Results

Thirty-eight patients with persistent secondary hypogammaglobulinemia as a consequence of anti-inflammatory treatment of their rheumatic disease have been identified and compared with 38 patients with PID, visiting our outpatient clinics at same time. Most of them (*n* = 32) had CVID, four had IgG deficiency falling under unclassified antibody deficiency and two had CID with reduced IgG and IgA levels. Secondary hypogammaglobulinemia appears to have a later onset than PID, as its diagnosis was made at a later age than PID ((53.42 y (interquartile range, IQR: 47.5–61.25) Vs. 38.5 y (IQR: 24.75–53.75)) (Table 1). On average, secondary hypogammaglobulinemia was diagnosed 7.3 y (IQR: 1–10.75) after the first diagnosis of rheumatic disease. Patients with secondary hypogammaglobulinemia had considerably higher levels of all measured Ig classes at diagnosis of hypogammaglobulinemia. There were 8/38 patients with secondary hypogammaglobulinemia and who had low IgA in addition to low IgG and 3/38 had low IgM. In contrast to the PID patients, who nearly all (37/38) were receiving immunoglobulin replacement treatment, only a minority of rheumatic patients with hypogammaglobulinemia required immunoglobulin replacement, which comes in accordance with the significantly higher IgG values in the case of secondary hypogammaglobulinemia. The majority of studied patients with secondary hypogammaglobulinemia had received more than one anti-inflammatory drug in addition to prednisolone, including DMARDs and biologics in combination with prednisolone (Table 2). Most rheumatic patients (34/38) were diagnosed with hypogammaglobulinemia while treated with anti-inflammatory regimens based on conventional DMARDs. There were 4/32 patients receiving oral prednisolone monotherapy and 13/38 developed hypogammaglobulinemia as a consequence of the first and only DMARD, which in most cases (11/13) was methotrexate. Most rheumatic patients displayed no significant disease activity at time measurement of peripheral lymphocyte subsets. SLE patients had a mean SLEDAI-2K score of 1.8 (IQR: 0.75–2.5). Except for two, all SLE patients had low-disease activity (SLEDAI-2K ≤ 2) considering the serologic parameters C3, C4 and anti–double-stranded DNA antibodies. The same holds true for the majority of studied patients with RA (9/11), who were in remission (DAS28-CRP ≤ 2.6, mean DAS28-CRP: 2.1, IQR: 1.6–2.6). Furthermore, nearly all studied patients (36/38) were receiving at time of testing no or a low-dosed prednisolone (≤5 mg/d, patient 11 was receiving 10 mg/d prednisolone and was on tapering and patient 13 had a maintenance dose of 7.5 mg/d).

Comparison of T cell counts in patients with PID and secondary hypogammaglobulinemia revealed a significantly higher percentage of memory CD4^+^ T cells and CD4^+^ T follicular cells in cases of primary hypogammaglobulinemia than in those with secondary (Figure 1). No substantial differences were observed for the rest of evaluated T cell subsets. This difference in proportion of memory CD4^+^ T cells and CD4^+^ T follicular cells among CD4^+^ T cells corresponded to increased absolute cell counts in the PID group as compared to secondary hypogammaglobulinemia (Appendix A). With respect to the B cell subsets, we found an increased proportion of naïve, transitional, plasma and CD21^low^ B cells in peripheral lymphocytes from PID patients as compared to patients with secondary hypogammaglobulinemia (Figure 2). In contrast, the proportions of memory/marginal zone B cells and cl. sw. memory B cells were significantly reduced in cases of PID. With the exception of memory/marginal B cells, absolute counts for B cell subsets corresponded with changes in their proportions among B cells (Appendix A). Significantly lower levels of all studied immunoglobulin classes in patients with primary hypogammaglobulinemia than those with secondary could be the consequence of differences in the composition of peripheral lymphocyte subsets. To test this, we evaluated if immunoglobulin levels associate with the proportions of lymphocytes subsets. This revealed significant association of IgG or IgA values with the percentages of cl. sw. memory B cells in both patients with primary and secondary hypogammaglobulinemia (Appendix A), suggesting that lower immunoglobulin values may be the consequence of alterations within the B cell compartment. However, both in case of PID and secondary hypogammaglobulinemia patients, T cell subset proportions did not associate with immunoglobulin levels (Appendix A). Six among the PID patients were receiving anti-inflammatory drugs at the time of evaluation of their lymphocyte subset counts (Appendix A). All six of them displayed reduced percentages of class-switched memory B cell counts and the majority had increased CD4^+^ T follicular cells, similar to the majority of PID patients, who were receiving no anti-inflammatory drugs. This suggests that changes in lymphocyte subsets in PID are rather related to disease-intrinsic mechanisms and are not the consequence of anti-inflammatory drugs, though the limited number of PID patients as well as the diversity of anti-inflammatory drugs does not allow firm conclusions concerning the way anti-inflammatory medications influence the lymphocyte phenotype.

Most studied patients with secondary hypogammaglobulinemia were diagnosed with either RA (11/38) or SLE (10/38). Despite the limited number of patients, separate analysis of lymphocyte subsets in patients with RA and secondary hypogammaglobulinemia revealed significantly reduced proportions of memory CD4^+^ and CD4^+^ follicular T cells and an increased percentage of class-switched memory B cells as compared with the PID patients (Figure 3). In case of SLE changes in memory, CD4^+^ and CD4^+^ follicular T cells did not reach significance, likely due to the small number of tested patients. However, within the B cells, SLE patients had significantly higher proportions of memory/marginal zone B cells and class-switched B and lower naïve B cells, compared to PID patients (Figure 3). As mentioned previously, the secondary hypogammaglobulinemia group included 11 patients, who each developed hypogammaglobulinemia as a consequence of methotrexate in combination with prednisolone. All those patients achieved remission, allowing prednisolone tapering to 0–5 mg/day. Aiming at identifying a likely drug-specific effect on peripheral T and B cell counts, we performed separate analysis of the relatively sizeable group of patients on methotrexate, which revealed similar differences to the ones described previously. In particular, increased proportions of CD4^+^ T follicular cells, memory CD4^+^ T cells as well as class-switched memory B cells and reduced naïve B cells were found in the methotrexate-treated patients as compared to the PID group (Appendix A). Differences for transitional and CD21^low^ B cells did not reach significance, likely due to the limited number of methotrexate treated patients.

Overall, lymphocyte phenotyping in patients with primary and secondary hypogammaglobulinemia identified a differential B and T cell subset distribution (Figure 4A). To evaluate the diagnostic accuracy of peripheral T and B cell subsets in discriminating primary from secondary hypogammaglobulinemia, values for each of the significantly changed parameters (i.e., the proportions of memory CD4^+^ T cells, CD4^+^ T follicular cells, naïve B cells, memory/marginal zone B cells, class-switched memory B cells, transitional B cells, IgM^+/-^ plasma cells and CD21^low^ B cells) were ranked, fitted into logistic regression and plotted into receiver operating characteristic (ROC) curves (Appendix A). Optimal threshold values for distinguishing PID from secondary hypogammaglobulinemia for each parameter and the corresponding area under the curve (AUC) values for each of the aforementioned parameters was calculated (Appendix A). The proportions of memory CD4^+^ T cells and CD4^+^ T follicular cells, as well as those of naïve B cells, class-switched memory B cells and CD21^low^ B cells, displayed adequate but still low sensitivity and specificity for the discrimination of primary from secondary hypogammaglobulinemia. Especially, the specificity values matching calculated threshold values were unsatisfactory, and were therefore adapted, to yield specificity higher than 85% (Table 3). Adapted threshold values and the respective proposed diagnostic criteria as well as their fulfilment by all studied patients with hypogammaglobulinemia are presented in Figure 4 (B and C). This adaptation of threshold values and the consequent increased specificity comes at the expense of a further reduction in sensitivity. Table 3 shows the diagnostic efficacy of the proposed diagnostic criteria in distinguishing primary from secondary hypogammaglobulinemia. The relatively poor performance of every single proposed criterion which can, however, be offset by an alternative consideration of both of the two better performing diagnostic parameters, which were the proportion of CD4^+^ T follicular cells and class-switched memory B cells. Alternative consideration of these two lymphocyte subsets displayed the best diagnostic efficacy among all other possible criteria combinations and could discriminate primary from secondary hypogammaglobulinemia with a specificity of 94.74% (c.i.: 88.25–99.36) and a sensitivity of 76.32% (59.76–88.56) (Table 3).

## 4. Discussion

A common side effect of anti-inflammatory regimens including corticosteroids, DMARD and biologics is hypogammaglobulinemia, which sometimes associates with clinically evident immunodeficiency [1,2]. In the present study approximately 20% of patients with secondary hypogammaglobulinemia displayed recurrent infections and were treated with immunoglobulin replacement. This—according to the literature—high proportion of patients on immunoglobulin replacement most likely reflects the fact that this study was conducted in a tertiary care center, where such patients are often referred to [30]. Furthermore, consistent with previous studies, suggesting the safety of most biologics with respect to the risk of infections [35,36,37], all studied rheumatic patients were diagnosed with hypogammaglobulinemia while treated with anti-inflammatory regimens containing conventional DMARDs.

PID and especially CVID may manifest as autoimmune diseases [6,7,8,9], which necessitates anti-inflammatory treatment with drugs that can result in hypogammaglobulinemia, independently of an underlying PID [2]. Measurement of immunoglobulin levels before starting an anti-inflammatory treatment, likely to cause hypogammaglobulinemia is therefore extremely important, as it may identify a preexisting hypogammaglobulinemia. The latter would be a per se primary hypogammaglobulinemia, falling under PID. Unfortunately, serum immunoglobulin levels before starting anti-inflammatory regiment are rarely tested by treating physicians, which may make the discrimination between primary and secondary hypogammaglobulinemia challenging. Furthermore, primary immunodeficiency and especially in CVID can be progressive and hypogammaglobulinemia may be absent in patients whose first manifestation is autoimmunity [38]. Our data suggest that hypogammaglobulinemia secondary to anti-inflammatory drugs differs from PID. In contrast to studied PID patients, the great majority of whom had combined IgG and IgA or IgM deficiency, the majority of patients with secondary hypogammaglobulinemia had an isolated IgG deficiency, which comes in accordance with a previous study on hypogammaglobulinemia in patients with giant cell arteritis or polymyalgia rheumatic [26]. However, approximately one fourth of studied patients with secondary hypogammaglobulinemia displayed a concomitant reduction of IgA and/or IgM, suggesting that isolated IgG deficiency cannot be a stringent criterion for discriminating primary from secondary hypogammaglobulinemia. Characterization of lymphocyte subsets among PID patients revealed abnormalities both within the B and the T cells. In case of B cells, in accordance with previous reports, we identify changes, such as the expansion of CD21^low^ B cells and the reduction of switched memory B cells that have been described in CVID [25]. With respect to the T cells, our data reveal expanded peripheral CD4^+^ T follicular helper cells, as described in other reports [29,39].

For rheumatic disease, on the other hand, a broad heterogeneity within B and T cell subsets has been described [40]. Furthermore, changes in lymphocyte subsets in rheumatic disease reflect not only disease-specific immune dysregulation but also the effect of anti-inflammatory regimens, which may be different for different drugs. Several studies are aimed at addressing the effect of single-drug regimens on peripheral lymphocyte subsets. However, discriminating the effect of single DMARD or biologics is difficult, as most patients had received glucocorticoids and/or other anti-inflammatory drugs. Glucocorticoids have been shown to affect lymphocyte trafficking and survival [41]. The selective decrease in transitional and naïve B cells in patients under glucocorticoid has been suggested to stem from increased apoptosis of human bone marrow progenitors [42]. Methotrexate can induce a reduction of transitional B cells, as well [43]. Within the T cells, methotrexate has been shown to induce an expansion of CD4^+^ T follicular helper cells. Several studies have evaluated the effect of biologics on peripheral lymphocyte subsets. For example, BAFF blockade in SLE affects non-memory B cells, inducing a reduction in transitional and naïve B cells [44]. Etanercept treatment specifically induced a reduction in IgM^+^ memory B cells in a cohort of patients with juvenile idiopathic arthritis (JIA) [43]. Besides B cell depletion, rituximab induced a depletion of CD4^+^ T cells in patients with RA, which was associated with clinical response [45]. Studies on peripheral lymphocyte subsets in rheumatic diseases such as rheumatoid arthritis, Sjögren’s syndrome, SLE and spondylarthritis have identified changes both within the T and B cells and in some cases associated those changes with disease activity or with phenotypic variations [46,47,48,49,50,51]. Overall, B and T cell subset abnormalities within rheumatic diseases, their association with disease activity or particular disease subgroups and/or anti-inflammatory drugs remain to be better elucidated in larger longitudinal cohorts including in the treatment of naïve patients.

Here, despite the heterogeneity of the secondary hypogammaglobulinemia group, including different rheumatic diseases and anti-inflammatory regimens, a comparison of lymphocyte subsets between patients with secondary hypogammaglobulinemia and those with PID revealed significant differences. Highly significant were the differences in the proportion of class-switched memory B and CD4^+^ T follicular T cells, which were matched to abnormally low class-switched memory and abnormally high CD4^+^ T follicular T cell percentages, suggesting the prominent disease-intrinsic effect on peripheral lymphocyte counts in PID. Separate analysis of patients with RA or SLE confirmed differences within T or B cell subsets, despite the limited number of studied patients. With respect to distinguishing primary from secondary hypogammaglobulinemia, considering changes in single cell subsets was of limited diagnostic efficacy. The proportion of class-switched memory cells and CD4^+^ T follicular cells displayed the best performance in terms of specificity, which however comes at the expense of low sensitivity. The latter could be considerably improved by collective consideration of class-switched memory and CD4^+^ T follicular cells.

In summary, peripheral lymphocyte subset counts and especially measurement of CD4^+^ T follicular helper cells and class-switched memory B cell counts could aid differentiation of primary from secondary hypogammaglobulinemia. Confirmation of our findings in additional cohorts of patients with hypogammaglobulinemia is needed to establish the diagnostic value of lymphocyte phenotyping in resolving the diagnostic dilemma between primary and secondary hypogammaglobulinemia. Studies addressing the effects of underlying rheumatic diseases and anti-inflammatory drugs on peripheral lymphocyte subsets could lead to the identification of disease- or drug-specific changes, which may further improve the diagnostic efficacy of T and B cell subset counts in differentiating PID from secondary hypogammaglobulinemia.

## Figures and Tables

**Figure 1 jcm-09-01049-f001:**
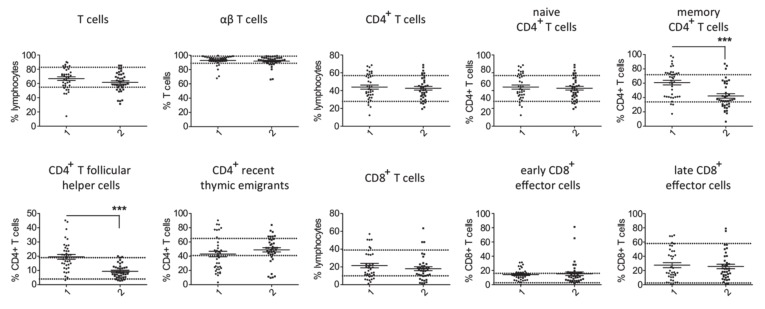
Comparison of T cell subsets in patients with primary (1, *n* = 38) and secondary (2, *n* = 38) hypogammaglobulinemia. Normal range values lie within the doted lines (*** *p* < 0.001).

**Figure 2 jcm-09-01049-f002:**
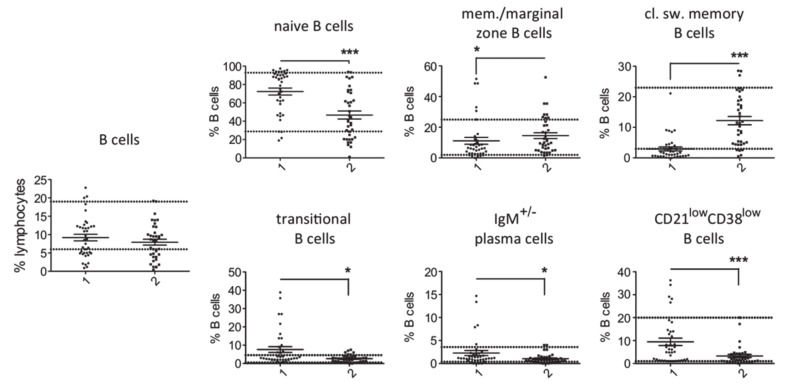
Comparison of B cell subsets in patients with primary (1, *n* = 38) and secondary (2, *n* = 37) hypogammaglobulinemia. Normal range values lie within the doted lines (* *p* < 0.05; *** *p* < 0.001).

**Figure 3 jcm-09-01049-f003:**
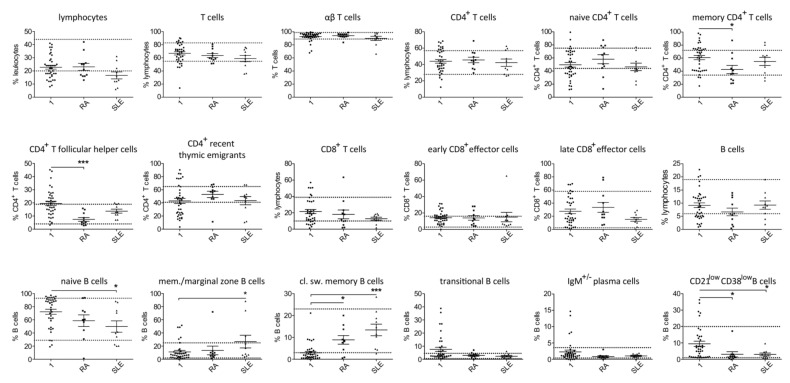
Comparison of T and B cell subsets in patients with primary (1) and secondary hypogammaglobulinemia with RA (*n* = 11 and *n* = 10 in case of B cell subsets, as a patient’s (pat. nr. 37) B cells could not be differentiated due to their very low count) and SLE (*n* = 10). (A) Normal range values lie within the doted lines (* *p* < 0.05; *** *p* < 0.001).

**Figure 4 jcm-09-01049-f004:**
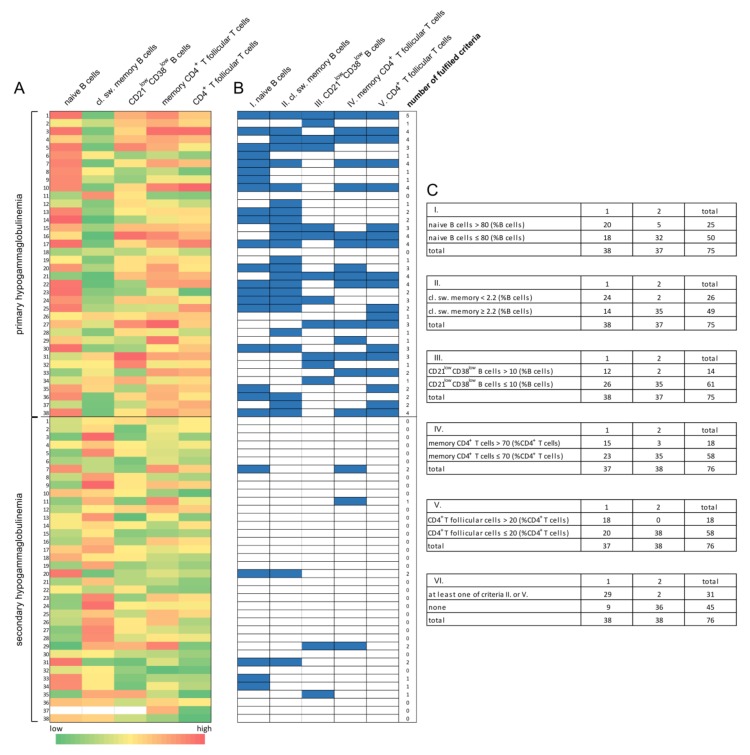
Diagnostic criteria for distinguishing primary from secondary hypogammaglobulinemia based in the differential B and T cell subset distribution in patients with primary immunodeficiency disorders (PID) as compared to patients with rheumatic disease and secondary hypogammaglobulinemia. (**A**) Heat map showing percentages of B and T cell subsets in all studied patients and highlighting their differential distribution. (**B**) Schematic representation of fulfilled diagnostic criteria (marked with blue) discriminating PID from secondary hypogammaglobulinemia. (**C**) Proposed diagnostic criteria for distinguishing primary from secondary hypogammaglobulinemia and cross tables showing all studied patients fulfilling or not fulfilling each time indicated criterion.

**Table 1 jcm-09-01049-t001:** Characteristics of studied patients with primary and secondary hypogammaglobulinemia.

	Primary Hypogammaglobulinemia (1)	Secondary Hypogammaglobulinemia (2)	*p* - Value
Number (*n*)	38	38	
Male Gender (*n* (%))	13 (34.21)	10 (26.32)	ns
Current Age (Mean (IQR))	51.82 y (37.5–60)	57.66 y (52–66.25)	ns
Age at Diagnosis of Hypogammaglobulinemia (Mean (IQR))	38.5 y (24.75–53.75)	53.42 y (47.5–61.25)	*** (<0.0001)
Patients Treated with Immunoglobulin Replacement (*n* (%))	37 (97.37)	7 (18.42)	*** (<0.0001)
IgG (Mean (IQR))	2.74 g/L (0.71–4.6)	5.63 g/L (4.99–6.22)	*** (<0.0001)
IgA (Mean (IQR))	0.45 g/L (0.04–0.48)	1.45 g/L (0.77–1.97)	*** (<0.0001)
IgM (Mean (IQR))	0.69 g/L (0.18–0.96)	0.83 g/L (0.52–1.05)	*** (0.0003)
Absolute Lymphocyte Count (Mean (IQR))	1653 (1019–2080)	1469 (1011–1838)	ns
Lymphocyte Percentage (Mean (IQR))	22.82 (16.75–28.25)	20.92 (13–27)	ns

IQR, interquartile range; ns, non-significant; y, years.

**Table 2 jcm-09-01049-t002:** Secondary hypogammaglobulinemia patients’ rheumatic disease, current and previous treatments. Prednisolone treatment is noted in case it is the only treatment.

Pat. Nr.	Rheumatic Disease	Current Therapy	Therapy at Diagnosis of Hypogammaglobulinemia	Previous Therapies
1	perSpA	MTX	MTX	
2	RA	MTX	MTX	
3	axSpA	adalimumab	MTX	MTX, SSZ
4	AOSD	prednisolone	prednisolone	
5	SSc	MTX	MTX	
6	perSpA	MTX + etanercept	MTX + SSZ	MTX, SSZ, LFN
7	SLE	MMF	CYC	CYC, HCQ
8	SLE	MTX	MTX	HCQ
9	SLE	MMF	MMF	
10	RA	MTX	MTX	HCQ, SSZ
11	SLE	CYC	CYC	AZA, MMF, HCQ
12	SLE	CYC	CYC	AZA, MMF, HCQ
13	RA	TCZ	MTX + RTX	MTX, LFN, SSZ, gold, adalimumab, infliximab
14	RA	MTX	MTX	
15	GCA	MTX	MTX	
16	RA	MTX + etanercept	MTX	MTX, HCQ
17	SLE	LFN + HCQ	MTX + HCQ	AZA, MTX, HCQ
18	axSpA	MTX + infliximab	MTX + infliximab	adalimumab
19	SS	prednisolone	prednisolone	
20	RA	MTX	MTX	
21	SS	MMF	MMF	CYC, AZA
22	PG	prednisolone	prednisolone	
23	AOSD	prednisolone	prednisolone	
24	SS	MTX	MTX	
25	SLE	MMF + HCQ	MMF	MMF
26	GCA	MTX	MTX	
27	SS	MTX	MTX	
28	SLE	AZA + HCQ	AZA + HCQ	MTX
29	RA	MTX	MTX	
30	RA	MTX	MTX	
31	RA	TCZ	SSZ + MTX	MTX, SSZ, LFN, adalimumab, abatacept
32	perSpA	adalimumab	SSZ + MTX	SSZ, MTX, HCQ
33	SLE	AZA	AZA	HCQ
34	SS	AZA + HCQ	AZA + HCQ	
35	SS	AZA	AZA	
36	SLE	MMF	MMF	HCQ
37	RA	MTX + RTX	MTX + RTX	LFN, MTX, HCQ, SSZ
38	RA	MTX + adalimumab	MTX + RTX	TCZ, etanercept, abatacept, SSZ, LFN, HCQ, MTX

AOSD, adult-onset Still’s disease; axSpA, axial spondyloarthritis; AZA, azathioprine; CYC, cyclophosphamide; GCA, giant cell arteritis; HCQ, hydroxychloroquine; LFN, leflunomide; MMF, mycophenolate mofetil; MTX, methotrexate; pat. nr., patient number; perSpA, peripheral spondyloarthritis; PG, pyoderma gangrenosum; RA, rheumatoid arthritis; RTX, rituximab; SLE, systemic lupus erythematosus; SS, Sjögren’s syndrome; SSc, systemic sclerosis; SSZ, sulfasalazine.

**Table 3 jcm-09-01049-t003:** Diagnostic efficacy of peripheral lymphocyte subset counts for differentiation of primary from secondary hypogammaglobulinemia.

Proposed Diagnostic Criterion	I. Naive B Cells >80 (%B Cells)	II. cl. sw. Memory B Cells <2.2 (%B Cells)	III. CD21^low^ CD38^low^ B Cells >10 (%B Cells)	IV. Memory CD4^+^ T Cells >70 (%CD4^+^ T Cells)	V. CD4^+^ T Follicular Cells >20 (%CD4^+^ T Cells)	VI. At Least One of Criteria II. or V
Sensitivity (%)	52.63	63.16	31.58	39.47	47.37	76.32
95% c.i.	35.82–69.02	45.99–78.19	17.5–48.65	24.04–56.61	30.98–64.18	59.76–88.56
Specificity (%)	86.49	94.59	94.59	92.11	100	94.74
95% c.i.	71.23–95.46	81.81–99.34	81.81–99.34	78.62–98.34	90.75–100.0	88.25–99.36
PPV	80	92.31	85.71	83.33	100	93.55
95% c.i.	59.3–93.17	74.87–99.05	57.19–98.22	58.58–96.42	81.47–100.0	78.58–99.21
NPV	64	71.43	57.38	60.34	65.52	80
95% c.i.	49.19–77.08	56.74–83.42	44.06–69.96	46.64–72.95	51.88–77.51	65.4–90.42
*p* - value	0.0005	<0.0001	0.0062	0.0024	<0.0001	<0.0001

c.i., confidence interval; PPV, positive predictive value; NPV, negative predictive value.

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
