# Peer review of "Peripheral Blood Lymphocyte Phenotype Differentiates Secondary Antibody Deficiency in Rheumatic Disease from Primary Antibody Deficiency"

_jcm, 2020, doi:10.3390/jcm9041049_

Round 1

Reviewer 1 Report

In this manuscript, Jablonka et al. have tried to develop a diagnostic tool to differentiate primary and secondary hypogammaglobulinemia by blood lymphocyte phenotyping using flow cytometry. By comparing B and T cell subsets from patients with PID and SID patients with rheumatic disease, who developed hypogammaglobulinemia, the authors observed significant differences in the proportion of several B and T cell subsets. Based on their results, the authors proposed criteria for the distinction of primary from secondary hypogammaglobulinemia. The results would help to establish the diagnostic value of lymphocyte phenotyping to distinguish primary and secondary hypogammaglobulinemia.

This manuscript was written in poor quality, in particular, of referring figures (Figure A3, 4, A4, A5, are all wrongly cited). Most figure citations seemed wrong, thus making it difficult to review it precisely. I do not think the authors have ever referred to figure 4 showing Diagnostic criteria for distinguishing primary from secondary hypogammaglobulinemia in the texts.

Since the manuscript measured several uncommon subsets of T and B cells such as early/late CD8+ effector cells, transitional B cell, Cl.sw.memory B cells, CD21low CD38low B cells, a brief introduction regarding the function and phenotype of the subsets would be nice.

As shown in Table 1, among the two groups of patients, there is a clear difference between 1’ and 2’ hypogamma patients. Thus, the suggested phenotypes could be associated with the antibody levels, not only with 2’ hypogamma patients. Patients with rheumatic diseases showed significant levels of IgG, but patients with hematological diseases usually showed much low levels of IgG, equivalent to 1’ hypogamma patients. It would be great to test the suggested immune phenotypes in 2’ hypogamma patients with hematological diseases.

The authors implied that the suggested immune phenotypes seem associated with anti-inflammatory treatment, not 2’ hypogamma itself. Even it is not a high proportion, 5% of 1’ immunodeficient patients with complications also receive anti-inflammatory treatment. Then would they fall into the 2’ hypogamma group? How does the author reconcile the risk?

The paragraph explaining table 3 was densely written. More elaboration is required. Can you draw a ROC curve with combining criteria II and V?

Author Response

With bold characters we indicate Reviewer’s comments and with an arrow (à) our answers.

This manuscript was written in poor quality, in particular, of referring figures (Figure A3, 4, A4, A5, are all wrongly cited). Most figure citations seemed wrong, thus making it difficult to review it precisely. I do not think the authors have ever referred to figure 4 showing Diagnostic criteria for distinguishing primary from secondary hypogammaglobulinemia in the texts.

à We are sorry for this mistake. Figures in Appendix have been renamed (from S1-S5 to A1-A5). In the last paragraph of the ‘Results’ section, ‘Figure 4’ was changed into ‘Figure 4A’ to better guide the reader through the three different parts of Figure 4 (now labeled as A, B and C). Wrongly referred ‘Figure 3’ in last paragraph of the ‘Results’ is now deleted and following sentence has been added, instead: ‘Adapted threshold values and the respective proposed diagnostic criteria as well as their fulfilment by all studied patients with hypogammaglobulinemia are presented in Figure 4 (B and C)’. To further improve the text we improved the language and separated text into more paragraphs in the ‘Discussion’ section.

Since the manuscript measured several uncommon subsets of T and B cells such as early/late CD8+ effector cells, transitional B cell, Cl.sw.memory B cells, CD21low CD38low B cells, a brief introduction regarding the function and phenotype of the subsets would be nice.

à We now provide a brief introduction on the function and phenotypes of measured lymphocyte subsets (see 3rd paragraph of the ‘Introduction’ section): ‘After activating antigen recognition, progressive differentiation of T cells can be traced by characterizing CD45RA, CD27 and CD28 expression [17]. Upon activation CD45RA switches to CD45RO, while CD28 and CD27 expression are lost sequentially in the course of T cell differentiation. In case of CD8+ T cell differentiation, CD27-CD28- cells (late effector cells) display an effector-like phenotype, whereas CD27+CD28- cells (early effector cells) appear to have a recent replicative history and partial effector function [18].  Coexpression of CD31 and CD45RA defines recent thymic emigrants, associating with the presence of T cell receptor excision circles (TRECs) [19]. Follicular T cells are antigen-experienced CD4+ T cells, expressing CXCR5. These cells regulate antigen-specific activation of B cells in the context of germinal center reaction but are also identified in peripheral blood [20]. After initial differentiation of B cells in the bone marrow, including the successful expression of a B cell receptor, transitional B cells emigrate from the bone marrow [21]. Expression of high levels of IgM and CD38 are used to characterize these cells. Naïve B cells are differentiated from memory ones on the basis of expression of CD27 and IgD [22]. CD21low B-cells are an innate-like memory B cell subset,  found increased in autoimmune diseases such as SLE and RA as well as in CVID patients with autoimmune manifestations [23-25]. Peripheral blood lymphocyte phenotyping has been traditionally employed to evaluate immunodeficiency within PID and define disease subgroups [25]. According to the current diagnostic criteria of CVID, characterization of particular B and T cell subsets is required for the diagnosis of CVID and its differentiation from CID [3]’.

As shown in Table 1, among the two groups of patients, there is a clear difference between 1’ and 2’ hypogamma patients. Thus, the suggested phenotypes could be associated with the antibody levels, not only with 2’ hypogamma patients. Patients with rheumatic diseases showed significant levels of IgG, but patients with hematological diseases usually showed much low levels of IgG, equivalent to 1’ hypogamma patients. It would be great to test the suggested immune phenotypes in 2’ hypogamma patients with hematological diseases.

à Indeed, antibody levels could associate with the phenotypes of peripheral lymphocytes. To test if the previously described significantly lower levels of all studied immunoglobulin classes in patients with PID associate with differences in lymphocytes subsets we performed a correlation analysis, which revealed significant association of lower IgG or IgA values with the percentages of cl. sw. memory B cells in both patients with primary and secondary hypogammaglobulinemia, suggesting that changes in antibody levels may be due to alterations within the B cell compartment. Both in case of PID and secondary hypogammaglobulinemia patients, T cell counts did not associate with immunoglobulin levels. Results of correlation analysis have been summarized in Table A1.

Regarding this finding we added following text in the ‘Results’ section (see 2nd paragraph): ‘Significantly lower levels of all studied immunoglobulin classes in patients with primary hypogammaglobulinemia than those with secondary could be the consequence of differences in the composition of peripheral lymphocyte subsets. To test this, we evaluated if immunoglobulin levels associate with the proportions of lymphocytes subsets. This revealed significant association of IgG or IgA values with the percentages of cl. sw. memory B cells in both patients with primary and secondary hypogammaglobulinemia (Table A1), suggesting that lower immunoglobulin values may be the consequence of alterations within the B cell compartment. However, both in case of PID and secondary hypogammaglobulinemia patients, T cell subset proportions did not associate with immunoglobulin levels (Table A1)’.

Unfortunately, we have no access to patients with hematological diseases and secondary hypogammaglobulinemia.

The authors implied that the suggested immune phenotypes seem associated with anti-inflammatory treatment, not 2’ hypogamma itself. Even it is not a high proportion, 5% of 1’ immunodeficient patients with complications also receive anti-inflammatory treatment. Then would they fall into the 2’ hypogamma group? How does the author reconcile the risk?

à 6 Patients (ID. 1, 7, 14, 20, 21 and 25) with PID, fulfilling the criteria of primary hypogammaglobulinemia were receiving anti-inflammatory drugs at time of evaluation of their lymphocyte subset counts. In ‘Appendix’ we have added ‘Table A2’, where we list those patients together with the indication of anti-inflammatory treatment and the drugs they were receiving at time of testing. All those patients had reduced percentages of cl.sw. memory B cell counts and the majority had increased CD4+ T follicular cells, fitting the values observed for the rest of PID patients, who have received no anti-inflammatory drugs. Despite the limited number of patients with PID receiving with anti-inflammatory drugs, this suggests that changes in lymphocyte subsets in PID are rather the consequence of disease-intrinsic pathomechanisms.

Considering this subgroup of PID patients, we have added the following text in the ‘Results’ section (see 2nd paragraph): ‘6 among PID patients were receiving anti-inflammatory drugs at time of evaluation of their lymphocyte subset counts (Table A2). All 6 of them displayed reduced percentages of class switched memory B cell counts and the majority had increased CD4+ T follicular cells, similar to the majority of PID patients, who were receiving no anti-inflammatory drugs. This suggests that changes in lymphocyte subsets in PID are rather related with disease-intrinsic mechanisms and not the consequence of anti-inflammatory drugs, though the limited number of PID patients as well as the diversity of anti-inflammatory drugs does not allow firm conclusions on the way anti-inflammatory medications influence lymphocyte phenotype’.

The paragraph explaining table 3 was densely written. More elaboration is required. Can you draw a ROC curve with combining criteria II and V?

à The paragraph explaining table 3 has been now rewritten to better present the data presented in table 3. The ROC for criterion VI (‘At least one of criteria II or V’) could not be drawn, as the input parameters would not be continued and the outcome of the all proposed criteria is dichotomous (fulfilled (positive)/not fulfilled (negative)) [1-3]. Alternatively, an ROC curve could be drawn by e.g. simple addition of the raw values of studied subjects after the conversion into a common scale, but this would not take into account the proposed threshold for each cell subset, and consequently, would not reflect criteria VI. We have instead calculated the sensitivity, specificity, PPV, and NPV for criterion VI, which represent our proposal and for all other combinations of criteria. The diagnostic performance of all other combinations was inferior to the proposed criterion 6. Data are not shown but inferiority, especially in terms of specificity is evident, as figure 4B depicts every single criterion and it fulfilment (marked with blue color) or not-fulfilment by every single studied subject.

References for this answer:

  1. Swets JA. ROC analysis applied to the evaluation of medical imaging techniques. Invest Radiol. 1979;14:109–21.
  2. Hanley JA. Receiver operating characteristic (ROC) methodology: the state of the art. Crit Rev Diagn Imaging. 1989;29:307–35.
  3. Karimollah Hajian-Tilaki. Receiver Operating Characteristic (ROC) Curve Analysis for Medical Diagnostic Test Evaluation. Caspian J Intern Med. 2013; 4(2): 627–635.

Reviewer 2 Report

The authors conducted a case-control study to examine the differences in peripheral B- and T- subpopulations due to immunosuppresive treatment with different anti-rheumatic drugs in comparison with primary immunodeficiency.

Your comparator population of secondary hypogammaglobulinemia patients consists of a very diverse group, with different diseases and therapeutic strategies.

Biologics are also included, and they do not pose a global immunosuppresive effect more often seen in traditional agents such as GCs, AZA, MTX. 

I do not understand the rationale behind the inclusion of different rheumatic diseases (of varying etiology, systemic inflammatory disease, mixed connective tissue disease and large vessel vasculitis) with diverse pathogenesis to be studied together, without sufficent sampels in each subgroup. Why are 2 GCA patients, 1 AOSD, 1 pyoderma included?

Please provide a rationale for why these patients are included. If you want to demonstrate reliability in heterogeneous rheumatic disease populations the sample size for constituent diseases has to be larger. Otherwise this is a dense stream of observations analyzed without a strict, prior research question in mind. 

Furthermore, prior history of medication use shows that some patients have been subjected to different therapeutic regimens, which could reflect a more refractory disease course. Since these diseases have different pathogenesis, and potentially even diverse immunophenotypes, the use of various agents can confound your findings. Steps should be undertaken to define a homogenous study population, or account for confounding factors. For example, in the setting of RA, use of methotrexate with RTX may even have a positive effect on infection susceptibility and hypogammaglublinemia. What about the influence of steroid dosage, disease activity and serologic status? This should be taken into account when describing a homogenous study population.

(Cobo-Ibáñez T, et al. Rheumatol Int. Boleto, G., et al. 2018, Seminars in Arthritis and Rheumatis)

Author Response

With bold characters we indicate Reviewer’s comments and with an arrow (à) our answers.

Biologics are also included, and they do not pose a global immunosuppresive effect more often seen in traditional agents such as GCs, AZA, MTX. 

à All studied patients had received traditional agents and indeed none developed hypogammaglobulinemia as a consequence of a monotherapy with biologics. Anti-inflammatory regimens at diagnosis of hypogammaglobulinemia are shown in Table 2. To better describe Table 2 we have added the following sentence (see ‘Results’ section, 1st paragraph): ‘Most rheumatic patients (34/38) were diagnosed with hypogammaglobulinemia while treated with anti-inflammatory regimens based on conventional DMARDs’. Further, we added following comment in the discussion (see ‘Discussion’ section, 1st paragraph): ‘Further, consistent with previous studies, suggesting the safety of most biologics with respect to the risk of infections [35-37], all studied rheumatic patients were diagnosed with hypogammaglobulinemia while treated with anti-inflammatory regimens containing conventional DMARDs’.

 I do not understand the rationale behind the inclusion of different rheumatic diseases (of varying etiology, systemic inflammatory disease, mixed connective tissue disease and large vessel vasculitis) with diverse pathogenesis to be studied together, without sufficent sampels in each subgroup. Why are 2 GCA patients, 1 AOSD, 1 pyoderma included?

à We have acknowledged the heterogeneity of the secondary hypogammaglobulinemia group at several points in the discussion, which indeed represents a limitation of our study. However, our goal was to evaluate if there are differences in lymphocyte subset counts between patients with primary and secondary hypogammaglobulinemia. Therefore, we had no reason to exclude AOSD or pyoderma patients, especially in that there is scarce if at all evidence on altered lymphocyte subsects for these rheumatic diseases. Analysis of collected data from this heterogeneous group of patients which secondary hypogammaglobulinemia revealed significant differences in lymphocytes subsets when compared to PID patients, the majority of whom had abnormally low cl. sw. memory B cells and abnormally high CD4+ T follicular T cells, whereas for nearly all rheumatic patients cells were within normal range. The diagnosis of underlying rheumatic disease could be considered for the relatively sizeable subgroup of patients with SLE and RA. Analysis of cell counts from these two subgroups yielded significant differences described in Figure 3.

Please provide a rationale for why these patients are included. If you want to demonstrate reliability in heterogeneous rheumatic disease populations the sample size for constituent diseases has to be larger. Otherwise this is a dense stream of observations analyzed without a strict, prior research question in mind. 

à Here we aimed at evaluating if peripheral blood lymphocyte phenotype differentiates primary antibody deficiency from secondary hypogammaglobulinemia, defined as hypogammaglobulinemia developing as a consequence of anti-inflammatory regimens. All recruited patients with secondary hypogammaglobulinemia had therefore normal levels of all major immunoglobulin classes prior to initiation of anti-inflammatory treatment. In contrast PID patients had received no prior anti-inflammatory treatment at diagnosis of hypogammaglobulinemia.

In other words, our rationale was to compare PID with non-PID patients. The PID group is more homogenous and the majority of patients had subnormal low cl. sw. memory B cells and subnormal high CD4+ T follicular cells, which was not the case of patients with secondary hypogammaglobulinemia, whose values remained in most cases within normal range (Figure 1 and Figure 2). Despite heterogeneity of the non-PID group esp. these two cells subsets can distinguish PID from non-PID as suggested by ROC (Figure A5) and analysis of performance of proposed diagnostic criteria (Table 3).

We do recognize that the underlying rheumatic disease, its activity as well as the anti-inflammatory treatment may affect lymphocyte subset counts. We therefore suggest in the discussion, that longitudinal studies including treatment naïve patients may aid clarifying the effect of the above stated parameters (see ‘Discussion’, last sentence of 3rd paragraph). Further, in the last paragraph of the ‘Discussion’ we state that ‘Studies addressing the effect of underlying rheumatic diseases and anti-inflammatory drugs on peripheral lymphocyte subsets could lead to the identification of disease- or drug-specific changes’. Finally, we conclude: ‘Studies addressing the effect of underlying rheumatic diseases and anti-inflammatory drugs on peripheral lymphocyte subsets could lead to the identification of disease- or drug-specific changes, which may further improve the diagnostic efficacy of T and B cell subset counts in differentiating PID from secondary hypogammaglobulinemia.

Observed and as suggested diagnostic differences presented in the present study are according to presented data the consequence of the more prominent changes in lymphocyte percentages in the PID group, which we comment in the 4th paragraph of the ‘Discussion’:  Highly significant were the differences in the proportion of class-switched memory B and CD4+ T follicular T cells, which were matched to abnormally low class-switched memory and abnormally high CD4+ T follicular T cell percentages, suggesting the prominent disease-intrinsic effect on peripheral lymphocyte counts in PID’.

Furthermore, prior history of medication use shows that some patients have been subjected to different therapeutic regimens, which could reflect a more refractory disease course. Since these diseases have different pathogenesis, and potentially even diverse immunophenotypes, the use of various agents can confound your findings. Steps should be undertaken to define a homogenous study population, or account for confounding factors. For example, in the setting of RA, use of methotrexate with RTX may even have a positive effect on infection susceptibility and hypogammaglublinemia. What about the influence of steroid dosage, disease activity and serologic status? This should be taken into account when describing a homogenous study population.

(Cobo-Ibáñez T, et al. Rheumatol Int. Boleto, G., et al. 2018, Seminars in Arthritis and Rheumatis)

à Disease activity, the required treatment and of course the disease itself, may all affect lymphocyte subset counts, as discussed in our previous answers, which needs to be further investigated. However, it is noticeable that abnormally high CD4+ T cell counts and abnormally low cl. sw. memory B cells are commonly observed among PID patients and have been identified in just a few patients with secondary hypogammaglobulinemia. Considering prominent abnormalities within the PID group, we would not expect that these parameters confound our data, which however needs to be shown in further studies as suggested.

Most studied rheumatic patients displayed no significant disease activity. SLE patients had a mean SLEDAI-2K score of 1.8 (IQR: 0.75-2.5). All except for two had low-disease activity (SLEDAI-2K≤2), considering the serologic parameters C3, C4 and anti–double‐stranded DNA antibodies. The same holds true for the majority of studied patients with RA (9/11), who were in remission (DAS28-CRP ≤2.6, mean DAS28-CRP: 2.1, IQR: 1.6 - 2.6). Further, nearly all studied patients (36/38) were receiving at time of testing a prednisolone dose ≤5 mg. Patient 11, who was receiving 10 mg/d prednisolone and was on tapering at time of testing and patient 13 had a maintenance dose of 5 mg/d. The limited number of patients with active disease allows no conclusion with respect to the effect of disease activity on measurement of peripheral lymphocyte subset counts. This data have been added in the first paragraph of the ‘Results’ section.

Steps should be undertaken to define a homogenous study population or account for confounding factors. This would necessitate considering all likely confounding factors, including underlying rheumatic disease, therapeutic regimen and disease activity.  We therefore  conclude (see last paragraph of ‘Discussion’): ‘Studies addressing the effect of underlying rheumatic diseases and anti-inflammatory drugs on peripheral lymphocyte subsets could lead to the identification of disease- or drug-specific changes, which may further improve the diagnostic efficacy of T and B cell subset counts in differentiating PID from secondary hypogammaglobulinemia.

Round 2

Reviewer 1 Report

The authors addressed most of my concerns. The manuscript has been much improved. 

I am surprised to find out that the authors did not cite all figures in texts, even though I have pointed it out in my review. 

Author Response

We apologize for not mentioning Figure 3, which was wrongly referred to as ‘Figure 4’ in text. We have now corrected this mistake and checked that all figures are appropriately cited in text (see below).

Figure

line

Figure 1

169

Figure 2

175

Figure 3

199, 203

Figure 4

214, 229

Figure A1

123

Figure A2

123

Figure A3

173

Figure A4

211

Figure A5

220

We would like to thank the Reviewer for his/her suggestions which improved quality of our manuscript.

Reviewer 2 Report

The authors responded extensively to my suggestions.

Author Response

We would like to thank the Reviewer for his/her suggestions which improved quality of our manuscript.
